

# Cword2vec: a novel morphological rule-based word embedding approach for Urdu text sentiment analysis

Saquib Khushhal[1], Abdul Majid[1], Syed Ali Abass[1], Rabia Riaz[1], Mohammad Babar[2] and Shafiq Ahmad[3]

[1] Department of Computer Science and Information Technology, University of Azad Jammu & Kashmir, Pakistan, Muzaffarabad, Pakistan
[2] Department of Computing and Electronics Engineering, Middle East College, Muscat, Oman
[3] Industrial Engineering Department, College of Engineering, King Saud University, Riyadh, Saudi Arabia

## ABSTRACT

Word embeddings are essential to natural language processing tasks because they contain a single word's syntactic and semantic information. Word embeddings have been developed widely for numerous spoken languages across the globe like English. The research community needs to pay more attention to the Urdu language despite its significant number of speakers, which amounts to approximately 231.3 million individuals. Urdu is a complex language because word boundaries in Urdu are unspecified, as it does not employ delimiters between words. The compound word, a multiword expression, is a more complex word consisting of many strings or independent base words. Traditionally, compound words are identified during the word segmentation using bigram or trigram approaches. The challenge with these techniques is that they do not produce meaningful words. This study uses morphological rule-based compound words in Urdu text documents. For text representation, a self-trained morphological rule-based compound word embedding (Cword2vec) based on the word2vec model is proposed for Urdu text sentiment analysis. The performance of self-trained morphological rule-based compound word embedding was then evaluated using four well-known deep learning models, *i.e.*, long short-term memory (LSTM), bidirectional LSTM (BiLSTM), convolutional neural networks (CNN), and convolutional LSTM (C-LSTM) for sentiment analysis. We also compare the performance of morphological rule-based compound words with traditional compound word identification techniques such as bigrams and trigrams. Regardless of the classification model, word embedding using our proposed morphological rule-based compound words outperformed in terms of precision, recall, F1 score, and accuracy than bigrams and trigrams.

# INTRODUCTION

Word embedding is an essential step in natural language processing, which involves transforming contextual information from a *corpus* into a lower-dimensional space

Corresponding authors
Saquib Khushhal,
saqib.khushhal@ajku.edu.pk
Abdul Majid, majid@ajku.edu.pk

characterized by real numbers (*Haider, 2018*; *Pham & Le, 2018*). The input layer of the approach is commonly used in many NLP applications, including but not limited to document development (*Fan, Lewis & Dauphin, 2018*), text or document simplification (*Chandrasekar, Doran & Bangalore, 1996*), document summarization (*Nenkova & McKeown, 2012*), next-word prediction (*Barman & Boruah, 2018*), and next-sentence prediction (*Shi & Demberg, 2019*). There are many word embedding approaches to represent words present in text documents. Many early researchers used a one-hot code word embedding approach to represent words in numeric form. This method involves taking the text and dividing it into individual tokens. For each word, a vector is generated with a size equal to the vocabulary. The issue with one-hot code approaches is that they only represent true or false words. If a word exists in a document, it shows 1; otherwise, it shows 0. The issue at hand can be resolved by employing embeddings to enhance efficacy through the utilization of a fully connected layer. The fully connected layer is referred to as the embedding layer, while the weights associated with it are known as embedding weights. In the subsequent stage, rather than performing the multiplication between the hidden layer and the one-hot vector layer, the weight values are extracted from the embedding weight matrix. This procedure involves multiplying a one-hot vector with an embedded weight matrix to obtain a row matrix representing the index of a specific input unit. Multiple strategies and methods exist for defining words in vector form, such as word2vec, GloVe, FastText, CBOW, and others. Various commonly spoken languages have adopted the latest word embedding techniques for language processing tasks. However, Urdu, the 10th most widely spoken language, has received relatively little attention in this regard.

Over the past decade, there has been a significant increase in the popularity of websites in many languages, including Urdu. Urdu has been designated an official language in Pakistan. A substantial global population widely uses it for communication. More research needs to be conducted on Urdu text sentiment analysis (*Syed, Aslam & Martinez-Enriquez, 2010*; *Mukhtar & Khan, 2018*). It is not feasible to apply techniques designed for languages such as English in the context of Urdu sentiment analysis. The Urdu language presents numerous challenges for sentiment analysis owing to its intricate morphological complexity. Compared to other languages, the current state of language processing resources needs to be improved in several aspects, including stop word lists, lemmatization, stemming tools, and tokenization or word segmentation.

Word segmentation plays a crucial role in various information retrieval and sentiment classification tasks by serving as a fundamental feature vector. It can break down and separate written text into meaningful units known as tokens. During tokenization, compound words, names, words with suffixes, and abbreviations must also have a single boundary (*Lehal, 2010*). Urdu word segmentation is challenging for several reasons, but the most common one is identifying compound words. The compound word, a multiword expression, is a more complex word consisting of many strings or independent base words. Mainly, the Urdu language uses a combination of two or more words to pronounce single words, *e.g.*, the word "خوش آمدید" (welcome).

Traditionally, the word segmentation process identifies compound words using bigram or trigram approaches. The challenge with these techniques is that they produce

**Table 1 Compound words identification during tokenization of sentence using bigram, trigram, and morphological compound words.**

| Bigram | Trigram | Compound words |
|---|---|---|
| آسمان و (the sky and) | آسمان و زمین (heaven and earth) | آسمان و زمین (heaven and earth) |
| و زمین (and the earth) | و زمین کی (and of the earth) | آسمان و زمین کی پیدائش (birth of heaven and earth) |
| غور و (consider) | غور و فکر (observation) | غور و فکر (observation) |
| و فکر (and worry) | فکر کی دعوت (an invitation to thought) | غور و فکر کی دعوت (an invitation to ponder) |
| کی دعوت (invitation to) | | |

meaningless words. Morphological knowledge of the language can be used to overcome these issues. This study uses a morphological rule-based approach to identify compound words for word segmentation. Table 1 provides the result of tokenization of the following Urdu sentences when bigram, trigram, and proposed morphological-based approaches are used. Consider a sentence:

”آسمان و زمین کی پیدائش کا ذکر اور ان میں غور و فکر کی دعوت قرآن پاک کی ہے“

In the above-given sentence, the compound words generated using the morphology of language are: ”آسمان و زمین“ (heaven and earth) ”آسمان و زمین کی پیدائش“ (birth of heaven and earth) ”غور و فکر“ (observation) and ”غور و فکر کی دعوت“ (an invitation to ponder). When the bigram method is used the compound words identified are ”آسمان و“ (the sky and), ”و زمین“ (and the earth), ”غور و“ (consider), and ”و فکر“ (and worry). The trigram will produce ”غور و فکر“ (heaven and earth) ”و زمین کی“, (and of the earth) ”آسمان و زمین“ (observation) ”فکر کی دعوت“, (an invitation to thought). The complete list of compound words generated by each method is given in Table 1. Trigram and bigram approaches identified meaningless compound words such as ”و زمین کی“ (and of the earth), ”و فکر“ (and worry) that hinder the understanding of textual data, whereas when morphological knowledge is used for compound word identification, the more meaningful compound words are produced such as ”آسمان و زمین“ (heaven and earth). This has motivated us to use morphological knowledge of Urdu to effectively identify compound words for the segmentation of textual data.

As discussed previously, word segmentation plays a crucial role in sentiment classification by serving as a fundamental feature vector. Sentiment analysis is an approach in natural language processing that analyzes user reviews of a service or product to determine whether they are positive (p), neutral (o), or negative (n). Sentiment analysis can be done at document, sentence, or aspect levels. Document-level sentiment analysis refers to evaluating a document's sentiment polarity (positive, neutral, or negative) within a collection of documents D. There may be a positive document with several negative sentences. The problem is solved in sentence-level classification, where every sentence is taken as a separate analytical unit, and it is a more detailed level of analysis. There are two

major tasks at this level: subjectivity classification and sentiment classification. Subjectivity classification identifies sentences as subjective (with an opinion) or objective. Sentiment classification is the categorization/classification of sentences/reviews as negative, positive, or neutral. Sentence-level categorization is challenging because of the Semantic Orientation of context-dependent words (*Mukhtar & Khan, 2018*). A text analysis method that classifies and characterizes the sentiment of a document related to its aspect. Aspect-based sentiment analysis is a computational task in natural language processing that requires determining and retrieving the sentiment related to features or components of a specific service or product.

Lexicon-based, machine learning, deep learning and are the three methods that are widely used for sentiment analysis. Lexicon-based methods use vocabulary and the sentiment associated with each word. We need a list of dictionaries called a lexicon to use a lexicon-based approach. These lexicons are related to their intensity and polarity. Machine learning techniques, on the other hand, rely on classification strategies to identify and label reviews as positive or negative. However, to train classifiers, this method needs labeled data. Numerous approaches, including both supervised and unsupervised methods, can be utilized for sentiment analysis. These methods have drawbacks, such as their limited adaptability to new data, their limited lexicons, and their inability to do sentiment analysis on large datasets due to their predetermined number of assignations.

Deep learning algorithms can perform perfectly for classification tasks with even the smallest datasets (*Barz & Denzler, 2020*; *Ramdan et al., 2020*). When dealing with extensive training datasets, deep learning models typically require more time for execution yet testing them after training generally consumes significantly less time. A recurrent neural network (RNN) is a deep learning method that may be applied to time series and sequential data using hidden layers. In RNN-based representation learning, the most well-known approaches include long short-term memory (LSTM), bidirectional LSTM (BiLSTM), and gated recurrent unit (GRU). While all the source information is compressed into a fixed-size vector, longer words can be challenging to manage in these types of networks. These neural networks can be used with attention mechanisms to improve performance (*Kardakis et al., 2021*). Text classification features of convolutional neural networks (CNN) have recently attracted a lot of media coverage. Using convolutional filters, CNN can identify recurring patterns in data. It has been shown that CNN models require 10 times less processing time than RNN models (*Hou et al., 2020*). For deep learning classification methods to learn data patterns, textual data must be translated to numerical form, known as word embedding. The choice of word representation can have an impact on the classification process. The probability of achieving accurate classification can be enhanced using an encoding method that effectively captures semantic relationships.

In this study, a self-trained morphological rule-based compound word embedding based on the skip-gram of the word2vec model is proposed for text representation. The

performance of the proposed compound word embedding is then evaluated for sentiment analysis using deep learning models.

The primary contributions of this study are outlined below:

- Morphological rules are defined for all derivation of Urdu words, facilitating compound word identification during the tokenization process of Urdu text documents.
- A novel word self-trained morphological rule-based compound word embedding (Cword2vec) is proposed for text representation, based on the skip-gram of the word2vec model.
- Common deep learning techniques are employed for sentiment analysis include LSTM, BiLSTM, CNN, and convolutional LSTM (C-LSTM).
- The performance of the proposed word embeddings is thoroughly evaluated, with results compared against bigrams and trigrams approaches for sentiment classification.

The present study is organized as follows: "Introduction" presents Introduction and problem statement. "Literature Review of the Urdu Language" presents a comprehensive review of the appropriate research. "Methodology" discusses implementing deep-learning sentiment analysis methods using morphological-based compound words as a feature vector and word embeddings. In "Experiments and Results", we present the findings and explore the implications of the suggested approach.

## LITERATURE REVIEW OF THE URDU LANGUAGE

### Word segmentation

Most natural language processing tasks require transforming syntactic features into real number vectors before transferring them to deep learning models. Word embedding solves the issue by developing vector representations of distributed words. There are multiple approaches and tools for producing word embeddings and dimension reduction, encompassing probability models (*Wiśniowski et al., 2020*), co-occurrence matrix, and knowledge-based models. One approach for word embedding is the one-hot coding embedding technique (*Pham & Le, 2018*). *Haider (2018)* was one of the early researchers who dedicated their efforts to studying Urdu and developed word feature vectors. The author employed the skip-gram model as a word embedding technique. Word embeddings of a compound word in Japanese were attempted to be composed of their constituent words, as described in *Komiya et al. (2023)*. Short and long units were employed, the basic building blocks of terms in UniDic, a Japanese dictionary compiled by the National Institute of Japanese Language and Linguistics. Word embeddings for compound words were trained using a *corpus* produced by merging separate corpora for their constituent words and their compound terms.

Urdu word embedding is the main emphasis of *Nazir et al. (2022)*. To accomplish this goal, they employed a dataset that included articles from several news outlets (including Entertainment, Sports, Business, Health, Politics, Science, the World, and more). In the process of tokenizing this dataset, 288,000,000 tokens were generated. In addition, they

used skip-gram, often known as the word2vec model, to create the word vectors. Dimensions of the vectors used in the embedding were restricted to 100, 200, 300, 400, 500, 128 and 256, and 512. The Wordsim-353 and Lexsim-999 datasets with annotations were used for the evaluation. Wordsim-353 and Lexsim-999 both had improved Spearman correlation coefficients of 0.66 and 0.439, respectively. The outcomes were deemed superior when compared to the current gold standard.

For opinion mining in Arabic text, including tweets, reviews, and news articles, *Altowayan & Tao (2016)* mainly uses word embeddings as the feature source. They gather a large Arabic *corpus* from various sources to study word representations. Gujarati, spoken in western India, is one example of a low-resource language for which (*Joshi, Koringa & Mitra, 2019*) have created word vectors. They also created a collection of analog test data to gauge the quality of their embeddings. Additionally, they compared the results to those of pre-trained Gujarati models.

For the Persian language, *Zahedi et al. (2018)* conducted a comprehensive evaluation of word embeddings using a collection of lexical semantics tasks, including analogies, idea categorization, and word semantic relatedness. The experiments show that FastText (sg) and Word2Vec(cbow) are the most effective models. Word2Vec CBOW, Skip-Gram, FastText, and GloVe are the typical word embedding models that (*Dlamini et al., 2021*) chose to train on the 10 million token isiZulu National *Corpus* (INC) to produce isiZulu word embeddings. *Salama, Youssef & Fahmy (2018)* looks at adding morphological annotations to the embedding model to improve Arabic word embedding. Using linear compositionality, they refine the obtained word vectors to their lemma forms, creating lemma-based embedding. Arabic analogies and subjectivity analysis are used to evaluate the performance of their model. The semi-supervised learning technique (*Cotterell & Schütze, 2019*) encourages the vectors to encode a word's morphology, *i.e.*, words near together in the embedded space share morphological properties.

To create morpheme embedding that aligns with the morphology of Bahasa, a byte pair embedding (BPE) approach was employed (*Amalia et al., 2021*). The researchers employed a simple methodology by applying a filtering process to the BPE segmentation result using a predetermined set of Bahasa morphemes. To address the intricacies of Arabic morphology, *Alkaoud & Syed (2020)* presents two embedding approaches that modify the tokenization process of conventional word embedding methods such as Word2Vec and contextual word embedding models like bidirectional encoder representations from transformers (BERT).

Word embedding models for Turkish and English are analyzed (*Yeşiltaş & Güngör, 2020*). The word2vec word embedding model was the main focus. They experimented with resizing and rotating context windows to boost word representation quality. The goal was to train models with higher accuracy without extending the training period by adjusting the orientation of the context window. Fast text embeddings and NGrams models are adapted by *Khalid et al. (2021)* to be trained on their own constructed *corpus*. Word similarity results were compared using these trained embeddings and state-of-the-art methods.

The purpose of *Wu, Zhao & Li (2020)* was to create a vector representation of English texts based on phrases. The findings obtained from the Enwiki, DBLP, and Yelp datasets demonstrate that the Phrase2Vec model, which was developed in their study, has superior performance compared to existing state-of-the-art phrase embedding models in both the similarity test and the analogical reasoning challenges. PETC has an F1-value index advantage of almost 4% compared to the baseline text categorization methods. Using phrase embeddings, *Mahata et al. (2018)* offer an unsupervised method (Key2Vec) for rating critical phrases from scholarly publications. Phrase-BERT draws on a large dataset of phrases in context mined from the Books3 *corpus* and a smaller dataset of various phrasal paraphrases generated automatically using a paraphrase generation model (*Wang, Thompson & Iyyer, 2021*).

In an effort to improve the precision of sentiment analysis (SA), *Sehar et al. (2023)* introduces a framework for sentiment analysis at the concept level. By gathering data from YouTube, an extensive Urdu language dataset was compiled, comprising a variety of discussions and evaluations pertaining to subjects including politics, movies, commercial products, and political affairs. In order to enhance the accuracy of polarity detection, language norms and deep neural networks (DNN) were integrated into the dataset. In order to perform sentiment analysis, the suggested framework utilises predetermined criteria to initiate the passage of sentiment from words to concepts by capitalising on the interdependencies between words in a sentence, which are determined by grammatical rules in the Urdu language.

Symmetries are integrated into the deep learning model and architecture by *Ahmed et al. (2023)* in numerous ways. First, it presents a novel meta-learning ensemble approach that combines basic machine learning with deep learning models utilizing two Urdu-specific meta-classifiers. The ensemble method combines inter- and intra-committee classifier predictions at two levels. A comparison of deep baseline classifier committees and the ensemble model is also done. The study's findings are expanded by comparing the suggested ensemble strategy's efficiency to more advanced ones. The proposed model reduces training complexity and overfitting.

Word embeddings for the Sinhala language, evaluated by *Lakmal et al. (2020)*. The three most popular word embedding models Word2Vec (in both its Skipgram and CBOW variants), FastText, and Glove are tested and compared using two distinct valuation strategies: There are two types of evaluation: intrinsic evaluation and extrinsic evaluation. Using their default and optimized parameter configurations evaluates the efficacy of different static embedding models, such as word2vec and fastText (*Lugli et al., 2022*). Additionally, *Ali, Missen & Husnain (2021)* examines contextual models, namely BERT and GPT-2, with different training levels, including a transfer learning technique that leverages the general Sanskrit *corpus*. The topic of discussion pertains to the building of embeddings.

## Deep learning for sentiment analysis

Deep learning methods have been explored for Urdu text classification in recent years. Event detection in Urdu text was initially presented by *Ali, Missen & Husnain (2021)*.

Popular deep learning (DL) models, such as CNN, RNN, and DNN, perform multiclass event classification. The unique hybrid deep learning model proposed by *Salur & Aydin (2020)* uses multiple word embeddings (Word2Vec, FastText, character-level embedding) and multiple deep learning methods (LSTM, GRU, BiLSTM, CNN).

To solve the issue of sentiment analysis, *Rehman et al. (2019)* proposes a hybrid model consisting of LSTM with an intense CNN model. They call it the hybrid CNN-LSTM model. They train basic word embeddings with the Word to Vector (Word2Vc) method. After that, embedding is done, and the suggested model integrates the features collected from the convolution and global max-pooling layers by considering their long-term dependencies. The accuracy of the suggested model is enhanced by employing dropout technology, normalizing, and a rectified linear unit.

Using a one-layer CNN architecture for local feature extraction and a two-layer LSTM for maintaining long-term dependencies, *Ombabi, Ouarda & Alimi (2020)* propose a unique deep-learning model for Arabic language sentiment analysis. An support vector machine (SVM) classifier is fed the feature maps learned by CNN and an LSTM to produce the final classification. The words in this model are embedded using the FastText approach. Using rule-based, machine learning (SVM, NB, Adabbost, MLP, LR, and RF), and deep learning (CNN-1D, LSTM, Bi-LSTM, GRU, and Bi-GRU) methods, *Khan et al. (2022)* creates an Urdu sentiment analyzer for manually annotated datasets to establish benchmark performance.

The main objectives of *Safder et al. (2021)* encompass two key aspects: first, creating a human-annotated dataset designed explicitly for investigating sentiment analysis in Urdu, and second, assessing the performance of state-of-the-art models. The models were subjected to binary and ternary classification tasks, employing a range of neural networks and heuristics such as LSTM, RCNN, rule-based, N-gram, SVM, and CNN. The proposed approach for Urdu text sentiment analysis (UTSA) is introduced by *Naqvi, Majid & Abbas (2021)*. It employs a range of vector word representations and deep learning techniques to examine textual data to identify emotional content. They examine and evaluate different deep learning approaches in the context of sentiment analysis to assess their effectiveness. The neural network architecture employed in their study encompasses LSTM, the attention-based bidirectional LSTM (BiLSTM-ATT), CNN, and C-LSTM.

The proposed model is built upon LSTM and CNN feature learners, as suggested by *Altaf et al. (2023)*. They utilized various baseline methods, including LSTM, Adaboost, XGboost, Random Forest, multilayer perceptron (MLP), SVM, decision tree, and KNN, in their analysis of the *corpus. Majeed et al. (2022)* utilizes the linguistic characteristics of the Urdu language to analyze sentiment at the sentence level. It also employs standard machine learning methods to classify idioms and proverbs. They create a dataset that includes idioms, proverbs, and sentences from the news domain. They then extract features based on part-of-speech tags, boolean values, and numeric values from the dataset after carefully analyzing the Urdu language.

*Nazir et al. (2024)* study introduces enhanced text normalization and tokenization methodologies for Urdu. The study proposes numerous regular expressions and rules for

Urdu text normalization, such as eliminating diacritics, normalizing individual letters, and separating digits. In word tokenization, fundamental properties are delineated and extracted for each character in the text. A machine learning model uses these rules to forecast spatial data and tokenize language. A significant outcome of the study is the creation of the largest human-annotated dataset in Urdu script, a practical contribution that could benefit the audience's work, covering five distinct areas.

*Shabbir & Majid (2024)* presents deep learning methodologies for sentence-level Urdu sentiment analysis (Urdu SA) within the context of MIoT. Their methodology comprises several phases: data collection, text preprocessing, model training, testing, and evaluation. A dataset of 25,000 Urdu reviews is utilised to train the proposed models. This dataset was constructed by extracting information from many Urdu blogs and social media sites, with a portion of the IMDB dataset utilised following its translation into Urdu. Native Urdu speakers annotate data, and several preprocessing techniques, such as tokenisation and stemming, are employed. This article trains two deep learning models, CNN and LSTM, on preprocessed Urdu reviews to ascertain their feelings. Both models undergo testing with diverse hyperparameter combinations, and each model's accuracy and F1 scores are assessed. The study results indicate that the LSTM model surpasses the CNN model, with a 96% accuracy and a 91% F1 score, paving the way for practical applications in Urdu sentiment analysis.

*Irum & Tahir (2025)* proposed a deep learning hybrid model that combines BiLSTM with a single layer multi-filter convolutional neural network (BiLSTM-SLMFCNN). The proposed and baseline methodologies are implemented on the Urdu Customer Support dataset and the IMDB Urdu movie review dataset. Pretrained Urdu word embeddings appropriate for sentiment analysis at the document level. The outcomes of different strategies are assessed, and our suggested model surpasses all other deep-learning methods for Urdu sentiment analysis. BiLSTM-SLMFCNN surpassed the baseline deep learning models, with accuracies of 83%, 79%, 83%, and 94% on small, medium, and big IMDB Urdu movie review datasets and the Urdu Customer Support dataset, respectively.

## METHODOLOGY

In this section, we delve into text representation, detailing data preprocessing, morphological rule-based compound word identification, and the proposed compound word embedding (CWord2Vec) centered on compound words. Subsequently, we elaborate on classification models utilized for sentiment classification of Urdu text documents. Figure 1 depicts the architecture of the methodology for sentiment classification of Urdu text documents using the proposed CWord2Vec embedding.

### Text representation

For natural language processing problems, text representation is an essential step in deep learning. Feeding text data into a deep learning model must first be converted into a numerical representation. One-Hot Encoding, term frequency-inverse document frequency (TF-IDF), Bag of Words (BoW), and word embeddings are all examples of

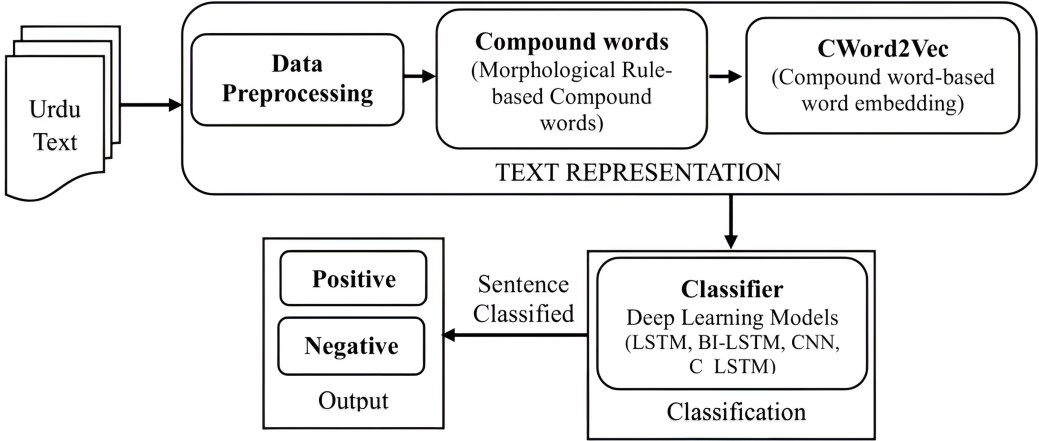

**Figure 1 Architecture of methodology for sentiment classification of Urdu text documents using proposed CWord2Vec embedding.**

popular text representation approaches used in deep learning for natural language processing.

In this study, text representation is achieved in three steps: (1) data preprocessing, (2) compound word identification using morphological rules, and (3) developing compound word embeddings (CWord2Vec) centered on compound words. All of these steps are elaborated in the subsequent subsections.

### Data preprocessing

Our data preprocessing phase comprises the following steps aimed at enhancing the performance of classification models for sentiment analysis. These steps are also illustrated with an example of an Urdu sentence in Fig. 2.

- The data is cleaned by eliminating alphanumeric characters, punctuation marks, and non-Urdu characters. Tokenization is then carried out to segment the text into words or tokens.
- Each word or token is tagged with a Part of Speech (POS) using Stanza (*Qi et al., 2020*). Stanza offers a suite of precise and effective tools for linguistically analyzing numerous human languages. Beginning with raw text, Stanza segments it into sentences and words. It can then identify parts of speech and entities.

### Morphological rules for compound word identification

The morphology of a language can serves as a basis for formulating rules to identify compound words. Table S1 provides a summary of the rules utilized for identifying compound words in Urdu. The table also shows the morphemes used during our morphological analyzer to extract compound words from word forms.

### Compound word embedding (Cword2vec)

The most common method for representing words in NLP is through one-hot encoding. In this approach, each term is represented by a sparse binary vector with a length equal to the

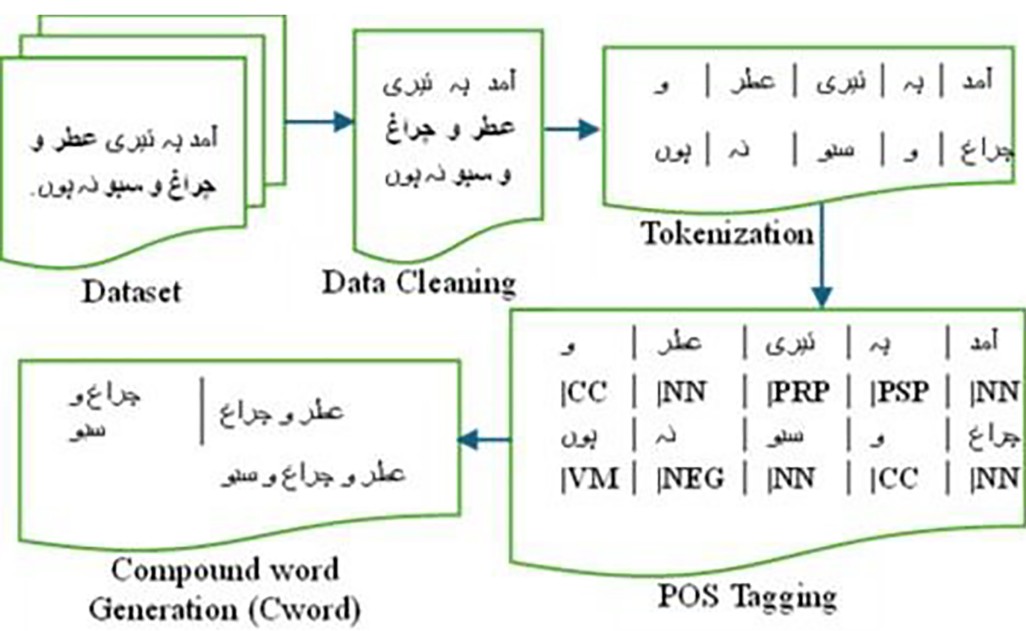

**Figure 2 Preprocessing steps for Urdu compound word generation.**

vocabulary size, where only one element is set to 1. The rest are 0, where 1 denotes the presence of that term in the vocabulary. However, this representation is typically inadequate for most deep-learning applications due to its high dimensionality and lack of semantic information.

However, the issues above are effectively addressed by word embeddings, which include the representation of words as dense vectors of a fixed length. The basic idea underlying word embeddings is that semantically connected words are represented by vectors positioned closely to one another in the vector space. These vectors' relative distances and placements reflect the semantic relationships between the words. Word2Vec, GloVe (Global Vectors for Word Representation), and FastText are well-known tools for generating word embeddings. In this study rather than only words/tokens we use compound word text representation.

In this study, a self-trained morphological rule-based compound word embedding (Cword2vec) based on the Word2Vec model is proposed for text representation. The Continuous Bag of Words (CBOW) and Skip-gram models are standard implementation techniques for word2vec embedding.

a) **Continuous Bag of Words (CBOW) model**

The CBOW model uses the words around the target word to predict the target word. The goal is to maximize the likelihood of predicting the target word $w_t$ from the context words $w_{t-c},\dots w_{t-1}, w_{t+1},\dots, w_{t+c}$. The number of words gives the context window size before and after the target word, denoted by C.

The following equations define the CBOW model:

1. **Input layer:** First, the Input Layer takes word embeddings representing context words ( $X_{t-c}$,...., $X_{t-1}$, $X_{t+1}$,.... $X_{t+c}$).

2. **Hidden layer:** For $i$ in [1, c], the average of the word embeddings is calculated as

$$h = \left(\frac{1}{(2*c)}\right) * sum(X_{t-i} + X_{t+i})$$

3. **Output layer:** A size $N \times V$ weight matrix calculates scores (logits) for each word in the vocabulary using the formula $z = h * W$.

4. **SoftMax function:** We use the SoftMax function to calculate the probability from the scores, which look like
$y = SoftMax(z)$. Doing so guarantees accurate probability in the results.

5. **Loss function:** To train the model, we compare the predicted probability $y$ to the genuine one-hot encoded representation of the target word, and the loss function is the cross-entropy between the two.

b) **Skip-gram model**

On the other hand, the Skip-gram model accepts the target word as input and endeavors to predict the surrounding context words. Skip-gram inverts the CBOW model's function. Table S2 shows source text with compound words as a training sample.
Mathematical expressions for the Skip-gram model look like this:

1. **Input layer:** The input word embeddings for the target word $X_t$

2. **Hidden layer:** There is none (only a line from the input to the output layer).

3. **Output layer:** The weight matrix $(N \times V)$ employed in CBOW is also utilized to calculate the scores (logits) for each word in the vocabulary, $z = X_t * W$.

4. **SoftMax function:** Use the SoftMax function to transform the scores into probabilities:
$y = SoftMax(z)$.

5. **Loss function:** A cross-entropy loss between the model's predicted probability y and the actual one-hot encoded representation of the context words is used to train the model.

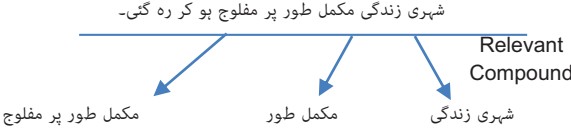

To implement Cword2vec, we initially detect compound words by employing a morphological rule-based method. Next, extract only the distinct compound words from the lexicon. In total, there are 97,162 unique compound words. A one-hot vector will be used to represent an input word, such as "شہری زندگی" (Urban Life). The vector will consist

of 97,162 components, with each component representing a word in our vocabulary. A value of "1" will be assigned to the position that corresponds to the compound word "شہری زندگی" (Urban Life) while all other positions will have a value of "0". The network's output is a singular vector consisting of 97,162 components. Each component represents the likelihood that a randomly chosen neighboring word corresponds to a compound word in our lexicon.

Figure S1 depicts the structural design of the skip-gram model. The buried layer does not employ an activation function, while the output neurons utilize SoftMax. Let us assume that we are acquiring knowledge about word vectors with 300 characteristics for this example. The hidden layer will be represented by a weight matrix consisting of 97,162 rows (corresponding to each word in our lexicon) and 300 columns (representing each hidden neuron).

The word vector for "شہری زندگی" (Urban Life) is then inputted into the output layer. The output layer functions as a SoftMax regression classifier. Every output neuron, representing a compound word in our lexicon, will generate an output ranging from 0 to 1. The total sum of these output values will always equal 1.

## Urdu text classification model

This section will discuss the deep learning models for sentiment classification of Urdu text documents. We built four distinct model representations using LSTM, BiLSTM-ATT, CNN, and C-LSTM. Except for CNN, our models use stacked layers because doing so produces a deeper network capable of making more precise predictions. The optimal combination of deep layers was finally established after several iterations.

### *Long short-term memory (LSTM)*

As depicted in Fig. S2, the suggested LSTM setup incorporates input layers through which parsed Urdu sentences are transmitted to the model. The subsequent two layers consist of stacked bidirectional LSTM layers, with 128 and 64 cells, respectively. After these layers, there is a dropout layer and a dense layer with two neurons and SoftMax activation. The final layer of the model is a fully connected output layer responsible for determining the polarity of the sentences received from the input layer.

In an LSTM cell, the calculations performed at time step t are as follows (mathematically):

(a) Compute the input gate, forget gate, and candidate values:

$$i_t = sigmoid(W_i * [h_{t-1}, x_t] + b_i) \tag{1}$$

$$f_t = sigmoid\big(W_f * [h_{t-1}, x_t] + b_f\big) \tag{2}$$

$$f_t = tanh\big(W_g * [h_{t-1}, x_t] + b_g\big) \tag{3}$$

(b) Update cell state:

$$c_t = f_t * [c_{t-1} + i_t] * g_t \tag{4}$$

(c) Calculate hidden and output gates:

$$h_t = o_t * \tanh(c_t) \tag{5}$$

$$o_t = sigmoid(W_o * [h_{t-1}, x_t] + b_o) \tag{6}$$

whereas,

- $x_t$ is input at time $t$.
- $h_{t-1}$ is the hidden state of previous time.
- $[h_{t-1}, x_t]$ is a symbol that joins $h_{t-1}$ with $x_t$.
- $W_i$, $W_f$, $W_g$, $W_o$ are weights and $b_i$, $b_f$, $b_g$, $b_o$ are bias vectors used to learn LSTM parameters.

### *Convolutional neural network (CNN)*

A convolutional neural network (CNN or ConvNet) is a type of deep learning model optimized for visual recognition applications. Because of their capacity to automatically learn and extract relevant information from images, CNNs have found great success in computer vision. They take cues from how the human brain processes visual information and are optimized for effectively navigating the spatial organization of images. Each CNN layer with an activation function is shown in Fig. S3. The main concepts and components of CNN are:

1. **Convolutional layers:** These layers use small filters/kernels to perform convolutional operations across the entire input image. Features like edges, textures, and forms can be more easily identified using convolutional filters. Many times, feature maps are what comes out of a convolutional layer.

2. **Activation functions:** A non-linear activation function (like ReLU - Rectified Linear Unit) is applied to each network element after each convolutional operation. With this, CNN can simulate complicated relationships between data points.

3. **Pooling layers:** These layers reduce the spatial dimensions of the feature maps, but the essential data is preserved. Max pooling, in which the highest value in a given region is chosen, and average pooling, in which the average value is determined, are two typical examples of pooling processes.

4. **Fully connected layers:** Following the several convolutional and pooling layers that make up a CNN, the last layer or layers are often fully linked. These layers, comparable to those in conventional neural networks, are responsible for making predictions using the information gained from the training process.

### *Hybrid CNN & LSTM (C-LSTM)*

The distinctive architecture of LSTM confers it a significant edge over conventional RNN models in extracting features and performing semantic analysis on long text sequences. In LSTM-based text classification tasks, there are two easy ways to extract the local characteristics from the outputs of LSTM: The two types of pooling commonly used in neural networks are max pooling and average pooling. The suggested model in this

research utilizes the output vector from the multi-layer LSTM model as the input vector for the CNN. It then constructs a CNN model on top of the multi-layer LSTM to extract the features of the input text sequences and enhance the classification accuracy.

Following the extraction of each feature sequence by the LSTM model, the resulting output is denoted as $H = [h1,\ h2,\ h3,\ \dots . ht]^T$. $h_t$ is the m-dimensional feature vector of the $t^{th}$ word in the text sequence. The magnitude of the vector is equivalent to the quantity of LSTM hidden layer nodes. $T$ is the number of LSTM expansion steps equivalent to the text sequence's length. The input matrix of the CNN is denoted as $H \in R^{m \times t}$. The convolution filter $F \in R^{j \times k}$, where j represents the number of words in the window, and $K$ is the dimension of the word embedding vector. The variable "$j$" represents the feature information of the $j$ adjacent words for every convolution operation. Step t's convolution filter $F = [F0,\ F1,\ F2,\ \dots . Fm-1]$ will provide a single value.

$$O_{F_i} = ReLU\left[\left(\sum_{i=0}^{m-1} h_{t+i}^T F_i\right) + b\right] \tag{7}$$

where $b$ is a bias, and $F$ and $b$ are the parameters of this individual filter. We employ the Rectified Linear Unit (RELU) as our activation function. The equation is as follows:

$$F(x) = \max(0, x). \tag{8}$$

### Bidirectional LSTM

When using a BiLSTM, signals can travel both in time and space. BiLSTM works by having two LSTM layers that process the input sequence in opposing directions (forward and backward, respectively). When constructing the final representation for a given time step, the hidden states of the two LSTM layers are combined, often using concatenation or summing. Visualization of Bidirectional LSTM is shown in Fig. S4.

In a BiLSTM, the calculations performed at time step $t$ are as follows (mathematically):

(1) **Forward LSTM (input-to-hidden):**

$$i_t = sigmoid(W_i * [h_{t-1}, x_t] + b_i) \tag{9}$$

$$f_t = sigmoid\left(W_f * [h_{t-1}, x_t] + b_f\right) \tag{10}$$

$$g_t = sigmoid\left(W_g * [h_{t-1}, x_t] + b_g\right) \tag{11}$$

$$o_t = sigmoid(W_o * [h_{t-1}, x_t] + b_o) \tag{12}$$

$$c_t = f_t * c_{t-1} + i_t * g_t \tag{13}$$

$$h_t = o_t * \tanh(c_t). \tag{14}$$

(2) **Backward LSTM (input-to-hidden):**

$$i_t' = sigmoid(W_i' * [h_{t-1}', x_t'] + b_i') \tag{15}$$

$$f_t' = sigmoid\left(W_f' * [h_{t-1}', x_t'] + b_f'\right) \tag{16}$$

$$g_t' = sigmoid\left(W_g' * [h_{t-1}', x_t'] + b_g'\right) \tag{17}$$

$$o_t' = sigmoid(W_o' * [h_{t-1}', \ x_t'] + b_o') \tag{18}$$

$$c_t' = f_t' * c_{t-1}' + i_t' * g_t' \tag{19}$$

$$h_t' = o_t' * \tanh(c_t') \tag{20}$$

(3) **Combining the hidden states from both directions:**

$$h_{t\_final} = [h_t, \ h_t'] \, (concatenation) \tag{21}$$

whereas,

- $x_t$ is input at time $t$.
- $h_{t-1}$ and $h_{t-1}'$ is the hidden state of the forward and backward LSTMs at the previous and next time steps, respectively.
- $[h_t, \ h_t']$ represents the concatenation of the hidden states from the forward and backward LSTMs.

## EXPERIMENTS AND RESULTS

This section describes the development environment, the dataset, compound word-based training, and evaluation measures used to assess the proposed compound word-based word embedding (CWord2Vec) for sentiment analysis. Additionally, it later discusses the results and provides a discussion.

### Development environment

Every experiment utilized the TensorFlow library with Keras within the Spyder 3.2.6 environment. Stanza was used as Part of the speech tagging (POS). For plotting, the Matplotlib Python library is used.

### Data

Blogs and news websites such as BBC, Express, DW, Dunya, and Humsub were used to collect data for this study, as reported in the study by *Naqvi, Majid & Abbas (2021)*. The dataset includes various topics, including politics, religion, Technology, Business, and literary studies.

The dataset is divided into Positive and Negative categories with 7,055 sentences. Statistics on the dataset are given in Table 2.

### Compound word (Cword2vec) training

The training parameters have been determined experimentally or by adopting the values of a previously trained model. Using various deep learning classifiers, we conducted many experiments on our dataset. Numerous experiments aim to identify an Urdu text dataset's most influential and precise binary classification model. To train any model, it needs to tokenize text into sub-words. This study uses morphological rule-based compound words during the tokenization process to train the model.

**Table 2 Data statistics for Urdu text sentiment analysis.**

| Type | Statistics |
| --- | --- |
| Number of positive reviews | 2,441 |
| Number of negative reviews | 2,049 |
| Total number of reviews | 7,055 |
| Total bigrams | 143,845 |
| Total trigrams | 139,358 |
| Total compound words (Morphological Based) | 97,162 |
| Minimum length of Bigrams | Min = 2 words |
| Maximum length of Bigrams | Max = 2 words |
| Minimum length of Trigrams | Min = 3 words |
| Maximum length of Trigrams | Max = 3 words |
| Minimum length of Compound words (Morphological Based) | Min = 2 words |
| Maximum length of Compound words (Morphological Based) | Max = 4 words |

The following are the hyperparameters assigned for model training:

- **Context window:** The selection of the number of words to be included on both sides of the input word within the *corpus* establishes input-output pairs.
- **Dimension:** The dimensionality of the word vector is to be acquired.
- **Learning rate:** The extent to which each training step influences weight is influenced.

Table 3 shows the parameter setting of different deep learning models used to evaluate Bigrams, Trigrams, and morphological rule-based compound words.

### Evaluation measure

The following evaluation measures are used to compare the performance of proposed morphological-based compound words with Bigrams and Trigrams.

$$Precision = \frac{TP}{TP + FP} \tag{22}$$

$$Recall = \frac{TP}{TP + FN} \tag{23}$$

$$F - Measure = 2 * \frac{Precion * Recall}{Precision + Recall} \tag{24}$$

$$Accuracy = 2 * \frac{TP + TN}{TP + TN + FP + FN}. \tag{25}$$

True positive, false positive, true negative, and false negative are represented by *TP, FP, TN*, and *FN*, respectively.

### Results and discussion

Using various deep learning classifiers, we conducted many experiments on our dataset. Numerous experiments aim to identify an Urdu text dataset's most influential and precise binary classification model.

**Table 3 Experimental parameters setting.**

| Training parameters | CNN | LSTM | BiLSTM | C_LSTM |
|---|---|---|---|---|
| Context window | 5 | 5 | 5 | 5 |
| Dimension | 300 | 300 | 300 | 300 |
| Learning rate | 0.01 | 0.01 | 0.01 | 0.01 |
| Dropout | 0.2 | 0.5 | 0.5 | 0.2 |
| Batch size | 64 | 64 | 64 | 64 |
| Activation function | 'ReLU' | 'ReLU' | 'ReLU' | 'ReLU' |
| Epochs | 21 | 21 | 21 | 21 |
| Optimizer | "Adam" | "Adam" | "Adam" | "Adam" |

This section compares traditional compound word identification using bigrams and trigrams with morphological-based compound words using deep learning classifiers. In this study, we use a morphological rule-based approach for word embeddings. This section discusses the performance evaluation of various deep learning (DL) models. In this work, we classify sentences using four deep learning models: LSTM, CNN, CNN with LSTM (C_LSTM), and Bidirectional LSTM. We extract three types of features for training and testing: morphologically based compound words, bigrams, and trigrams. The dataset distribution ratios are 80% and 20% for training and testing, respectively.

We employed the Wilcoxon Signed-Rank Test to ascertain the statistical significance of performance disparities among word embeddings. The assessments were conducted on classification scores derived from a diverse range of embeddings across a wide variety of datasets. A significance level of 0.05 ($p < 0.05$) was employed to ascertain the statistical significance of the observed differences. The findings validate that the proposed CWORD2VEC embedding surpasses alternative embeddings in sentiment categorization tasks.

The findings for the different models are compared in Table 4.

The best model results for each word embedding are indicated in bold. Models using compound word embeddings performed better than Bigram or Trigram embeddings in precision, recall, F1 score, and accuracy. When word embedding using compound words was used, all models performed better in accuracy, F1 score, precision, and recall. This table shows that the accuracy of deep learning algorithms lies between (61% to 68%) and (61% to 69%) for bigrams and trigrams, respectively.

In comparison, this accuracy lies between (69% to 75%) for compound words. Models based on morphological-based compound word embedding outperformed all others by achieving the highest precision, recall, F1 score, and accuracy. Unlike previous studies, our experiments show that C_LSTM is helpful for classification like other languages. Based on the results, it can be determined that C_LSTM performed better for Urdu text sentence-level sentiment classification regardless of embedding. It proved to be the preferred model for Urdu text classification. Regardless of the classification model, word embedding using our proposed morphological-based compound words outperformed in terms of precision, recall, F1 score, and accuracy than bigrams and trigrams. These

**Table 4 Comparative analysis of DL models embedding-wise.**

| Algorithm | Bigram Word2Vec | | | | Trigram Word2Vec | | | | Compound Word2Vec (CWord2Vec) | | | |
|---|---|---|---|---|---|---|---|---|---|---|---|---|
| | Pre | Re | F-1 | Acc | Pre | Re | F-1 | Acc | Pre | Re | F-1 | Acc |
| LSTM | 0.62 | 0.60 | 0.61 | 0.61 | 0.69 | 0.58 | 0.67 | 0.62 | 0.68 | 0.76 | 0.72 | 0.71 |
| BILSTM | 0.61 | 0.82 | 0.71 | 0.57 | 0.61 | 0.82 | 0.70 | 0.61 | 0.69 | 0.85 | 0.72 | 0.72 |
| CNN | 0.63 | 0.74 | 0.68 | 0.67 | 0.64 | 0.75 | 0.69 | 0.66 | 0.65 | 0.79 | 0.71 | 0.69 |
| C_LSTM | 0.61 | 0.81 | 0.70 | 0.68 | 0.64 | 0.81 | 0.71 | 0.69 | 0.71 | 0.84 | 0.75 | 0.75 |

observations show that morphological rule-based compound word embedding will be helpful for Urdu text sentence-level classification.

This study proposed a self-trained morphological compound word-based word embedding model for Urdu text sentiment analysis. In this study, we define our word embedding using the word2vec approach. This study compares word embedding using proposed morphological compound words with bigrams and trigrams embedding.

This study uses our self-trained (morphological based compound words) models to evaluate four well-known deep-learning approaches for classification. We assess the performance of our self-trained model, which is based on morphological compound words, by comparing it to the classic bigrams and trigrams model. In the course of the experimental process, it was seen that all models converged to yield identical outcomes following the completion of 21 epochs. Our suggested morphological-based compound word embedding improved accuracy, F1 score, precision, and recall across all models. Word embedding using compound words enhanced the deep learning model's performance compared to word embedding using bigrams and trigrams. Figure 3 shows the AUC score for each DL methodology that utilizes different word embeddings (*i.e.*, morphological-based compound word embedding, bigrams, and trigram embedding). It is studied that the ROC using word embedding with compound words performed consistently better for all deep-learning models. The word embedding performance of all three models, including morphological compound words, bigrams, and trigrams, achieved an AUC score of over 0.61. This level of performance is considered satisfactory for our applied classification technique. The AUC score reveals that all models with compound words-based word embedding had an AUC score greater than 0.69, and all models with word embedding using bigrams and trigrams had scores close to or above 0.66. Based on the analysis, every model effectively scaled the maximum number of sentences while employing word embedding derived from morphological rule-based compound word-based embedding.

This investigation faced many challenges due to several factors. It is possible that trying out several DL models and word embeddings on a small dataset failed to bring out the most outstanding results. However, considering that a smaller dataset is utilised to train the compound word embedding model, they are encouraging findings. A more comprehensive understanding may be attained by analysing a large dataset, which enables a more effective evaluation of the effectiveness of the self-trained compound word-based embedding learning paradigm in producing outstanding results.

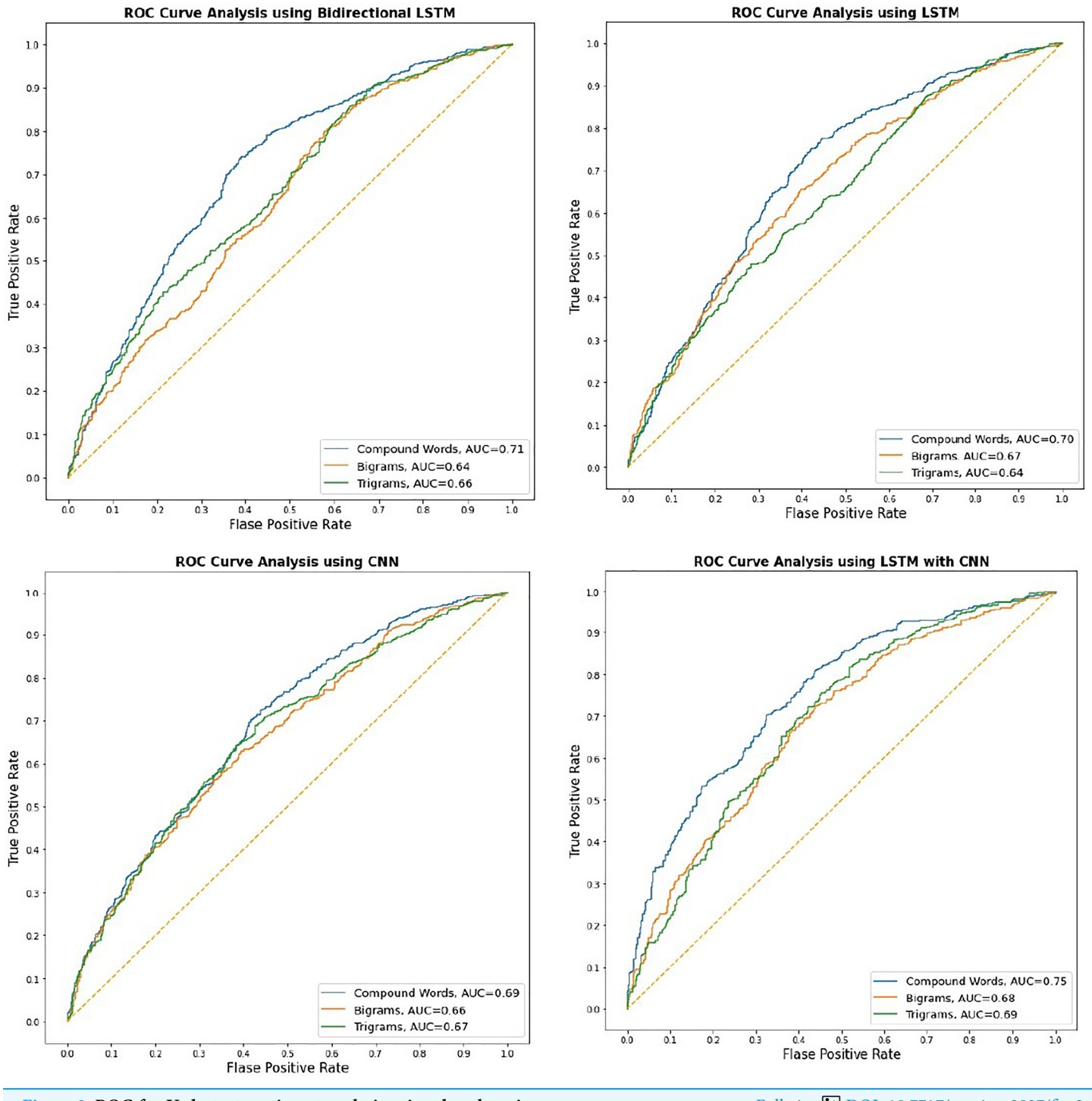

**Figure 3 ROC for Urdu text sentiment analysis using deep learning.**

## DISCUSSION

The results presented in the previous section demonstrate that morphological-based Compound Word performs better in terms of the Urdu text sentiment

analysis. The analysis performed in this subsection elucidates how using morphologically based compound words significantly increases the performance of Urdu text sentiment analysis.

The model performance evaluation involved analyzing misclassified samples from various embedding deployments. The analysis in Table 5 demonstrates the superiority of CWord2Vec over bigram and trigram Word2Vec embeddings in interpreting negative and context-based sentiment accurately. The experimental findings reveal that CWord2Vec successfully maintains compound words, further solidifying its position as the model of choice for sentiment analysis. To analyze the performance of morphological-based compound words, a few examples from the test data that are passed as input, along with their actual polarities, are classified by using compound words (C.W) and classified using bigram and trigram approaches as positive and negative sentences are shown in Table 5.

Table 5 shows three examples of Urdu text classified by our proposed compound word sentiment analyzer. In example 1 of Table 5, "ناجائز تجاوزات" (Illegal encroachments), "ناقص انتظامات" (Poor arrangements), "نکاسی آب" (Drainage) are compound words which are identified by using the morphological rule. See example 2; this sentence is morphologically positive, but if we use bigram or trigram, the classifier will classify the sentence as negative. In this sentence, two compound words change the morphology of the sentence; these compounds words are: "مصیبتوں کا مداوا" (Cure for troubles) and "اپنی مدد آپ" (help yourself). In this example alone, "مصیبتوں" (troubles) is negative but when it is combined with another noun "مداوا" (cure) using Noun-Izafat-Noun compound then this compound word considers as positive sentiment. So, this compound changes the behavior of the whole sentence. Moreover, in example 2 of Table 5 alone, "بود" and "باش" have no meaning, but when these words are combined by using an Inflectional compound (مرکب عطفی), this compound word gives us meaningful result "بود و باش" (style). As in example 3 of Table 5, "سونے" (gold) and "سہاگہ" (Sahaga) have neutral words, but when these words are combined using (مرکّب اضافی), now this shows compound word "سونے پر سہاگہ" (cherry on the cake). In this sentence, two compound words change the morphology of the sentence; these compounds words are: "مصیبتوں کا مداوا" (Cure for troubles) and "اپنی مدد آپ" (help yourself). in this example alone, "مصیبتوں" (troubles) is negative but when it is combined with another noun "مداوا" (cure) using the Noun-Izafat-Noun compound then this compound word considers as positive sentiment. So, this compound changes the behavior of the whole sentence.

Compound words have proven to be robust in sentiment analysis, particularly when a model is evaluated on a dataset different from the training dataset (*e.g.*, social media *vs.* news or academic texts). The robustness of transformers is further enhanced in mixed-language texts, such as Urdu-English code-switching, where n-gram approaches often struggle to understand meaningful contextual relationships. This robustness helps retain the semantic meaning of multi-word expressions, providing better feature representation and classification performance. Moreover, compound words alleviate data sparsity and enhance generalization among various linguistic dynamics, further highlighting their robustness on rich datasets. As such, compound words represent a

**Table 5  Example of sentences classified by Cword2vec (morphological compound word based) model of Urdu text.**

| Polarity with C. W | Polarity with bigram | Polarity with trigram | Ground truth polarity | Sentences |
|---|---|---|---|---|
| N | P | P | N | نکاسی آب کے ناقص انتظامات، برساتی نالوں میں کچرا اور نا جائز تجاوزات کی بھر مار کے باعث شہر کا انفرا اسٹرکچر چند گھنٹوں کی بارش کا بوجہ نہ اٹھا سکا۔ <br> (The inadequacy of water drainage systems, coupled with litter and illegal encroachments in the rainy channels, led to the city's infrastructure being overwhelmed, unable to bear the burden of heavy rainfall for several hours.) |
| P | Neu | Neu | P | اب طرز تعمیر اور انداز بود و باش کی باری تھی۔ <br> (Now it was the turn of architecture and style.) |
| P | N | Neu | P | علی نوکری ملنے پر خوش تھا اس پر سونے پر سہاگہ کہ اس کی تنخواہ میں بھی اضافہ ہو گیا! <br> (Ali was over the moon when she got the job - the cherry on top was a salary increase!) |

robust and promising target for many NLP tasks, providing reassurance in their effectiveness.

# CONCLUSION

This study proposed a self-trained morphological compound word-based word embedding using the word2vec model for Urdu text sentiment analysis. This study compares word embedding using proposed morphological compound words with Bigrams and Trigrams embedding. For sentiment analysis, the morphological rule-based compound word embedding performance was evaluated using different deep learning methods such as LSTM, BiLSTM, CNN, and C-LSTM. Each model's accuracy, F1 score, precision, and recall were computed using three different word embeddings (bigrams, trigrams, and morphological rule-based model). Unlike previous studies, our experiments show that C_LSTM is helpful for classification like other languages. Based on the results, it can be determined that C_LSTM performed better for Urdu text sentence-level sentiment classification regardless of embedding. It proved to be the preferred model for Urdu text classification. Irrespective of the classification model, word embedding using our proposed morphological rule-based compound words outperformed in terms of precision, recall, F1 score, and accuracy than bigrams and trigrams. From the above observations, morphological rule-based compound word embeddings proved to be better than traditional word embedding approaches like Bigrams and Trigrams in the performance assessment of different deep learning models for sentiment analysis.

In the future, deep learning models with more embeddings that provide better contextual information, and the increased and balanced dataset will be explored. Additionally, the extensive dataset can enhance accuracy, and morphological rule-based compound words can be employed to evaluate data translated from English to Urdu. Furthermore, to explore the performance of newer deep learning techniques such as transformer-based language models such BERT for classification needs to be explored.

## Limitation of the study

The lack of data and imbalance in the evaluation dataset limit our investigation. Our dataset has 7,055 utterances mixed with 2,565 positive and 4,490 negative samples. Due to

the dataset imbalance, deep learning models may behave differently and learn from the majority class. Although the dataset is small, our compound word-based segmentation algorithm works. However, using a larger and more diverse dataset can considerably improve the resilience of our word embeddings, increasing their generalization potential for sentiment analysis tasks and giving our research an optimistic future.

A large dataset can improve deep learning models by better representing speech patterns and moods like sarcasm, irony, and ambiguity. An extended dataset would help generate more equitable training data to reduce biases and assess CWORD2VEC embedding performance. Future research must prioritize a larger dataset with different linguistic patterns. It is important to validate and develop our methods, highlighting the need for other industry data.

### Funding
This research has received funding from the Center for Research and Consultancy, Middle East College Muscat, Oman The funders had no role in study design, data collection and analysis, decision to publish, or preparation of the manuscript.

### Grant Disclosures
The following grant information was disclosed by the authors:
Center for Research and Consultancy, Middle East College Muscat, Oman.

### Competing Interests
The authors declare that they have no competing interests.

### Author Contributions
- Saquib Khushhal conceived and designed the experiments, performed the experiments, performed the computation work, authored or reviewed drafts of the article, and approved the final draft.
- Abdul Majid conceived and designed the experiments, performed the experiments, analyzed the data, performed the computation work, prepared figures and/or tables, authored or reviewed drafts of the article, and approved the final draft.
- Syed Ali Abass conceived and designed the experiments, analyzed the data, performed the computation work, authored or reviewed drafts of the article, and approved the final draft.
- Rabia Riaz conceived and designed the experiments, performed the experiments, performed the computation work, authored or reviewed drafts of the article, and approved the final draft.
- Mohammad Babar conceived and designed the experiments, prepared figures and/or tables, and approved the final draft.
- Shafiq Ahmad conceived and designed the experiments, analyzed the data, authored or reviewed drafts of the article, provide funding for this work, and approved the final draft.

## Data Availability

The code and dataset are available in the Supplemental Files.

## Supplemental Information

Supplemental information for this article can be found online at http://dx.doi.org/10.7717/peerj-cs.2937#supplemental-information.

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
