# Peer review of "Cword2vec: a novel morphological rule-based word embedding approach for Urdu text sentiment analysis"

_PeerJ Computer Science, doi:10.7717/peerj-cs.2937_

## Round 0.1 · original submission · Major Revisions

Dear Authors,

Reviewers have now commented on your article. We do encourage you to address the concerns and criticisms of the reviewers with respect to reporting, experimental design, and validity of the findings and resubmit your article once you have updated it accordingly. Furthermore, all equations should be used with correct equation number. Explanation of the equations should be checked. Definitions and boundaries of all variables should be provided. Necessary references should also be given.

Best wishes,

Reviewer 1 ·

Basic reporting

1- A summary table can be provided in the "Literature Review" section.
In this way, the previous works can be understanded easily.

2- What are running time (execution time) of the methods?
Additional results (a new table or chart/graph) may also be given in terms of running times.

3- The organization of the paper (the structure of the manuscript) may be written at the end of the "Introduction" section.
For example: "Section 2 presents ... Section 3 gives ...."

4- The symbols in the text should be italic.
For example:
Line 450 "input at time t"
Line 404 "The weight matrix (N x V)"
Line 492 "information of the j adjacent words"
Line 493 "convolution filter F ="
Line 494 "Where b is a bias, and F and b are the parameters"

5- What is/are the limitation(s) of the study/method? An explanation can be added to the article.

6- In the reference list, there is no any paper published in 2024.
I suggest the authors citing the most recent papers.

7- A concern is that no formal statistical analysis of the results are done, to indicate whether the differences in performance are statistically significant or not.
For example; Friedman Aligned Rank Test, Wilcoxon Test, Quade Test, etc.
p-value can be calculated and compared with the significance level (p-value < 0.05).

8- It is lack of comparison. It can be compared with the results of the previous studies (existing studies in the literature).

Experimental design

5- What is/are the limitation(s) of the study/method? An explanation can be added to the article.

Validity of the findings

7- A concern is that no formal statistical analysis of the results are done, to indicate whether the differences in performance are statistically significant or not.
For example; Friedman Aligned Rank Test, Wilcoxon Test, Quade Test, etc.
p-value can be calculated and compared with the significance level (p-value < 0.05).

8- It is lack of comparison. It can be compared with the results of the previous studies (existing studies in the literature).

Additional comments

1- A summary table can be provided in the "Literature Review" section.
In this way, the previous works can be understanded easily.

2- What are running time (execution time) of the methods?
Additional results (a new table or chart/graph) may also be given in terms of running times.

3- The organization of the paper (the structure of the manuscript) may be written at the end of the "Introduction" section.
For example: "Section 2 presents ... Section 3 gives ...."

4- The symbols in the text should be italic.
For example:
Line 450 "input at time t"
Line 404 "The weight matrix (N x V)"
Line 492 "information of the j adjacent words"
Line 493 "convolution filter F ="
Line 494 "Where b is a bias, and F and b are the parameters"

5- What is/are the limitation(s) of the study/method? An explanation can be added to the article.

6- In the reference list, there is no any paper published in 2024.
I suggest the authors citing the most recent papers.

7- A concern is that no formal statistical analysis of the results are done, to indicate whether the differences in performance are statistically significant or not.
For example; Friedman Aligned Rank Test, Wilcoxon Test, Quade Test, etc.
p-value can be calculated and compared with the significance level (p-value < 0.05).

8- It is lack of comparison. It can be compared with the results of the previous studies (existing studies in the literature).

Reviewer 2 ·

Basic reporting

- The manuscript is well-written in professional and unambiguous English. However, there are minor grammatical issues and some awkward sentence constructions that could benefit from a language review by a native English speaker or a professional editing service.
- The paper provides a comprehensive literature review on Urdu text processing, word embeddings, and sentiment analysis. The authors reference relevant prior work, including related methods such as Word2Vec, GloVe, and FastText. The coverage of word segmentation in Urdu and the unique challenges it presents is thorough. However, it would be beneficial to include more recent citations on transformer-based models like BERT, as these are increasingly used in NLP tasks.
- The paper follows a conventional structure with clearly defined sections. The figures and tables are well-labeled and relevant. However, Figure 3, which depicts the architecture of the proposed model, could be more detailed, particularly in showing how the Cword2Vec embeddings interact with deep learning classifiers.
- The paper is well-structured and provides a clear motivation for the research. The hypothesis that a morphological rule-based compound word embedding improves sentiment classification in Urdu is clearly articulated and tested.

Experimental design

- The study is original and falls within the scope of PeerJ Computer Science. The research question is well-defined and addresses a knowledge gap in Urdu NLP. The motivation for developing a morphology-based word embedding for sentiment analysis is justified.
- The methodology is well-detailed, including: (1) Morphological rule-based compound word identification; (2) The proposed Cword2Vec embedding model based on Word2Vec (Skip-gram and CBOW); (3) Evaluation using deep learning models (LSTM, BiLSTM, CNN, and C-LSTM). However, the description of morphological rules used for compound word identification could be more explicitly detailed with examples. Whether these rules were manually crafted or derived automatically from a corpus is unclear.

Validity of the findings

- The dataset consists of 7055 sentences, which is relatively small for deep learning tasks. The paper does not mention whether data augmentation techniques were applied. Also, a discussion on potential overfitting should be included, particularly considering that multiple deep learning models are trained on a small dataset.
- The study convincingly demonstrates that Cword2Vec embeddings outperform traditional bigram and trigram-based embeddings across various deep learning classifiers. However, comparisons with other modern NLP embeddings, such as transformer-based embeddings (BERT, XLM-R, or fastText subword-based embeddings), are missing.
- The conclusions are well-supported by the results. However, a dedicated section discussing the limitations of the proposed approach is lacking. The following aspects should be addressed:
1) How does the proposed method perform on larger datasets?
2) How do results compare when tested on different domains (e.g., social media, news, or academic texts)?
3) How robust is the approach when applied to mixed-language text (Urdu-English code-switching)?

Additional comments

- The authors should consider adding a computational complexity analysis of their approach compared to other word embedding models.
- The paper's discussion section should include more on real-world applicability, such as deployment in sentiment analysis systems or integration into existing NLP frameworks.
- Since deep learning models are sensitive to hyperparameters, more details on hyperparameter tuning strategies (e.g., learning rates, dropout rates) would be useful.
- The dataset details should be made more explicit, particularly concerning preprocessing steps, tokenization, and train-test split criteria.

·

Basic reporting

“Cword2vec: a novel morphological rule-based word
embedding approach for urdu text sentiment analysis”
Main Revisions
Clarification of Methodology:
• The paper introduces a novel morphological rule-based approach for Urdu text sentiment analysis, but the methodology section could be more detailed. Specifically, defining and applying morphological rules for compound word identification needs further elaboration.
• Suggestion: Provide more examples of how morphological rules are applied to different types of Urdu compound words. Include a flowchart or pseudocode to make the process clearer.
Dataset Description:
• The dataset used for training and evaluation is described briefly, but more details are needed regarding its size, diversity, and how it was annotated.
• Suggestion: Include a table summarizing the dataset's statistics (e.g., number of sentences, distribution of positive/negative sentiments, sources of data). Also, discuss any potential biases in the dataset and how they were mitigated.
Comparison with State-of-the-Art:
• The paper compares the proposed approach with traditional bigram and trigram methods, but it lacks a comparison with more recent state-of-the-art methods for Urdu sentiment analysis, such as transformer-based models (e.g., BERT, GPT).
• Suggestion: Include a comparison with at least one transformer-based model to demonstrate the superiority or limitations of the proposed approach.
Evaluation Metrics:
• While the paper reports precision, recall, F1 score, and accuracy, it does not discuss other relevant metrics such as AUC-ROC, especially for imbalanced datasets.
• Suggestion: Include AUC-ROC curves and discuss the model's performance in terms of handling imbalanced data.
Error Analysis:
• The paper lacks a detailed error analysis to understand where the model fails and why. This is crucial for improving the model and understanding its limitations.
• Suggestion: Provide a qualitative analysis of misclassified examples and discuss potential reasons for these errors.
Reproducibility:
• The paper does not provide sufficient details for reproducibility, such as hyperparameters, training details, and code availability.
• Suggestion: Include a table with all hyperparameters used for training the models and consider making the code and dataset publicly available.
Minor Revisions
Introduction and Background:
• The introduction provides a good overview of the problem, but it could benefit from a more detailed discussion of the challenges specific to Urdu sentiment analysis.
• Suggestion: Expand the introduction to include more context about Urdu's linguistic complexity and why it poses unique challenges for sentiment analysis.

Literature Review:
• The literature review is comprehensive but could be better organized. Some sections feel disjointed, and the connection between the reviewed works and the proposed approach could be clearer.
• Suggestion: Reorganize the literature review to group related works together and explicitly state how the proposed approach builds on or differs from previous work.
Figures and Tables:
• Some figures (e.g., Figure 1, Figure 3) are not very clear and could be improved for better readability.
• Suggestion: Redesign the figures to make them more visually appealing and easier to understand. Ensure that all axes and labels are clearly marked.
Language and Grammar:
• While the language is generally clear, there are some grammatical errors and awkward phrasings that could be improved.
• Suggestion: Perform a thorough proofreading or consider professional editing to improve the language and grammar.
Conclusion:
• The conclusion is somewhat brief and does not fully discuss the broader implications of the findings or future work.
• Suggestion: Expand the conclusion to discuss the potential impact of the proposed approach on Urdu NLP and suggest specific directions for future research.
References:
• Some references are not formatted consistently, and a few key papers in the field of Urdu sentiment analysis are missing.
• Suggestion: Ensure that all references follow the journal's formatting guidelines and include recent, relevant papers in the field.
Additional Suggestions
• Ethical Considerations: If the dataset includes user-generated content (e.g., social media posts), the authors should discuss any ethical considerations, such as data anonymization and user consent.
• Broader Impact: Discuss the potential societal impact of the proposed approach, especially in regions where Urdu is widely spoken.
Summary of Recommendations:
1. Major Revisions:
• Clarify the methodology, especially the morphological rule application.
• Provide more details about the dataset and its annotation process.
• Compare the proposed approach with state-of-the-art models.
• Include additional evaluation metrics like AUC-ROC.
• Conduct a detailed error analysis.
• Ensure reproducibility by providing hyperparameters and code.

2. Minor Revisions:
• Expand the introduction and literature review.
• Improve the clarity of figures and tables.
• Proofread for language and grammar.
• Expand the conclusion.

Experimental design

.

Validity of the findings

.

Additional comments

.

---

## Round 0.2 · accepted · Accept

Dear Authors,

One of the reviewers did not respond to the invitation to review the revised manuscript. One reviewer accepted the invitation but did not send his/her review. One reviewer accepts your paper in its current form. Due to time constraints, I also have evaluated the revision myself and believe that your manuscript appears improved and ready for publication.

Best wishes,

Reviewer 1 ·

Basic reporting

The authors revised the manuscript adequately according to the reviewers' comments.
The manuscript is now more qualified and clear.
I have no further comments.

Experimental design

-

Validity of the findings

-

Additional comments

-